# The Effect of Intravenous Dexamethasone and Dexmedetomidine on Analgesia Duration of Supraclavicular Brachial Plexus Block: A Randomized, Four-Arm, Triple-Blinded, Placebo-Controlled Trial

**DOI:** 10.3390/jpm11121267

**Published:** 2021-12-01

**Authors:** Boohwi Hong, Chahyun Oh, Yumin Jo, Woosuk Chung, Eunhye Park, Hanmi Park, Seokhwa Yoon

**Affiliations:** 1Department of Anesthesiology and Pain Medicine, Chungnam National University Hospital, 282 Munhwa-ro, Jung-gu, Daejeon 35015, Korea; koho0127@gmail.com (B.H.); ohchahyun@gmail.com (C.O.); lemonny87@naver.com (Y.J.); woosuk119@gmail.com (W.C.); beareunbear@gmail.com (E.P.); invent2424@naver.com (H.P.); 2Department of Anesthesiology and Pain Medicine, College of Medicine, Chungnam National University, 266 Munhwa-ro, Jung-gu, Daejeon 35015, Korea; 3Big Data Center, Biomedical Research Institute, Chungnam National University Hospital, 282 Munhwa-ro, Jung-gu, Daejeon 35015, Korea

**Keywords:** dexmedetomidine, dexamethasone, intravenous adjuvant, regional anesthesia, brachial plexus block, postoperative analgesia

## Abstract

Intravenous dexamethasone and dexmedetomidine, in conjunction with peripheral nerve blockade, have each been reported to prolong the duration of analgesia. This study tested whether combined use further prolongs analgesia duration after supraclavicular brachial plexus block (BPB) in patients undergoing orthopedic upper extremity surgery. One hundred twenty patients were randomized 1:1:1:1 to Control (saline bolus and midazolam infusion [0.05 mg/kg loading, 20 µg/kg/h thereafter]); DMED (saline bolus and dexmedetomidine infusion [1 μg/kg loading, 0.4 μg/kg/h thereafter]); DEXA (dexamethasone [10 mg] bolus and midazolam infusion); and DMED-DEXA (dexmedetomidine infusion and dexamethasone bolus) groups. The primary outcome was the duration of postoperative analgesia, defined as the time from the end of the BPB to the first dose of analgesia via a patient-controlled device. Median (interquartile range) times to first dose of analgesia in the Control, DMED, DEXA, and DMED-DEXA groups were 8.1 (6.2–11.6), 9.0 (8.1–11.3), 10.7 (8.1–20.5), and 13.2 (11.5–19.1) hours, respectively (*p* < 0.001). Pairwise comparisons showed significant prolongation of analgesia in the DEXA included groups compared with the non-DEXA included groups (DEXA vs. control, *p* = 0.045; DEXA vs. DMED, *p* = 0.045; DMED-DEXA vs. control, *p* < 0.001; DMED-DEXA vs. DMED, *p* < 0.001). A mixed effect model showed that dexamethasone was the only significant factor for the prolongation of analgesia (*p* < 0.001). Intravenous dexamethasone prolonged the analgesia duration of supraclavicular BPB after orthopedic upper extremity surgery. The concurrent use of mild to moderate sedation dose of intravenous dexmedetomidine in addition to intravenous dexamethasone showed no additional benefit to the prolongation of analgesia.

## 1. Introduction

Brachial plexus block (BPB) is widely used for upper extremity surgery because it provides both intraoperative anesthesia and postoperative analgesia [1,2]. However, as the duration of analgesia is limited to the day of surgery [3,4], it may cause rebound pain, which can result in sleep disturbance and/or unplanned use of health care resources [5,6,7].

Prolonging the effect of BPB is of interest in regional anesthesia. Continuous regional analgesia using a catheter or intravenous or intraneural adjunctive agents such as dexamethasone [8] and dexmedetomidine [9,10,11] is a frequent option. Administration of continuous regional analgesia, however, is technically cumbersome and not suitable for ambulatory anesthesia in some countries, making adjunctive agents more attractive. Intravenous adjunctive agents are free of concerns regarding “off label use” and have been shown to be as effective as perineural injection [12,13,14,15,16].

According to our previous study, sedation with dexmedetomidine can prolong the analgesia duration of BPB [11]. We hypothesized that the prolonged analgesic duration of BPB induced by dexmedetomidine can be augmented by concurrent administration of intravenous dexamethasone. The effects of these adjunctive agents on the analgesia duration of BPB were therefore evaluated in a four parallel-arm study, in which patients received dexamethasone, dexmedetomidine, neither or both.

## 2. Materials and Methods

This study included 120 patients aged 20–70 years with American Society of Anesthesiologists (ASA) physical status I or II, who were scheduled for elective upper extremity surgery under supraclavicular BPB. Patients were excluded if they refused to participate; had preexisting neuropathy of the surgical limb, hypersensitivity to amide anesthetic, significant pulmonary disease, coagulopathy, sepsis, infection at the block site, or pregnancy. The study protocol was approved by the Institutional Research Board of Chungnam National University Hospital, Daejeon, Republic of Korea (CNUH 2019-04-022-002), and all participants provided written informed consent. The study was registered with Clinical Research information Service (CRIS, https://cris.nih.go.kr, KCT0004093, last accessed on 21 September 2021) on 26 June 2019, participants enrolled from 28 June 2019 to 26 June 2020.

Study data were collected and managed using REDCap (Research Electronic Data Capture) software hosted at Chungnam National University Hospital. REDCap is a secure, web-based platform designed to support capture of data for research studies [17]. This manuscript adheres to the applicable CONSORT (Consolidated Standards of Reporting Trials) guidelines [18].

Patients were randomized 1:1:1:1 to four groups: a Control group, a dexmedetomidine (DMED) group, a dexamethasone (DEXA) group, and a dexmedetomidine plus dexamethasone (DMED-DEXA) group. A computer-generated (www.randomization.com, accessed on 30 May 2019) randomization table with blocks of 4 and 8 was created. The sequence was uploaded into REDCap (redcap.cnuh.co.kr, accessed on 30 May 2019) to conceal the allocation, and a researcher (Y.J.) allocated each patient to the indicated group immediately after patient arrival at the operating room. After preparing the study drugs, the researcher exposed to the allocation was not involved in any part of study conduction. Other individuals who participated in the surgery, including the attending anesthesiologist, surgeon, and nurse, were blinded to the assignment.

### 2.1. Preparation of Study Drugs

Midazolam was used as a sedative agent in the control and DEXA group. Sedative agents included 30 mL of 4 µg/mL dexmedetomidine (Precedex Premix, Pfizer Pharmaceuticals Korea, Seoul, Korea) and 30 mL of 0.2 mg/mL midazolam, consisting of 6 mL midazolam (Midazolam inj, 1 mg/mL, ^®^Bukwang Pharmaceutical Co. Ltd., Seoul, Korea) added to 24 mL of normal saline, with 2 mL of 5 mg/mL dexamethasone (dexamethasone disodium phosphate injection^®^, Yuhan Pharm, Seoul, Korea) or normal saline included as a bolus intravenous adjunctive agent or placebo, respectively (Figure 1). All sedative agents were administered at a loading rate of 1.5 mL/kg/h for 10 min (i.e., 0.25 mL/kg for 10 min) and a maintenance rate of 0.1 mL/kg/h until the beginning of skin suture. These regimens translate into a 1.0 µg/kg loading dose and a 0.4 µg/kg/h continuous dose for dexmedetomidine and a 0.05 mg/kg loading dose and a 20 µg/kg/h continuous dose for midazolam. Drug dosages were based on ideal body weight.

### 2.2. Anesthetic Procedures

Standard ASA monitoring was applied before performing the block and was maintained throughout the entire procedure. All supraclavicular blocks were performed under ultrasound guidance by a single experienced anesthesiologist (B.H.) using an in-plane technique with a high-resolution ultrasound system (X-Porte, FUJIFILM SonoSite, Inc., Bothell, WA, USA), a high-frequency linear probe (HFL50xp: 15–6 MHz, X-Porte) and a nerve stimulator (0.1 ms, 0.5 mA, 2 Hz, sentinel mode, MultiStim SENSOR, PAJUNK, Geisingen, Germany). All patients were administered 25 mL of local anesthetic, consisting of a 1:1 mixture of 1% lidocaine and 0.75% ropivacaine. After confirmation of surgical readiness by pinprick in terminal nerve dermatomes (i.e., radial, median, ulnar, musculocutaneous) related to the operating field, the study drug was infused. A block failure was defined as the requirement for general anesthesia or additional infiltration of local anesthetics in the field due to inadequate anesthesia in the operating arm. Prior to sedation, supplemental oxygen was administered at a rate of 5 L/min via a simple facial mask. Once oral intake was tolerated in postoperative day, all patients received oral analgesia twice a day (combination of tramadol 37.5 mg and acetaminophen 325 mg, combination of naproxen 500 mg and esomeprazole 20 mg). Patient-controlled analgesia (PCA) devices (Accumate^®^1200, Woo Young Meditech, Seoul, Korea) were set to administer bolus doses of fentanyl 0.5 μg/kg (no continuous dose, lockout time of 10 min, total fentanyl dose of 1000 μg) and continued on discharge day. Rescue analgesic (intravenous pethidine 25 mg) was prescribed when the patient complaint pain greater than NRS 3 despite the use of PCA. All patients remained in the hospital for one to two days after surgery and were followed up at the outpatient clinic on 7 to 14 days after surgery.

### 2.3. Outcomes

The primary outcome was the time until the patient first required analgesia, defined as the time from the end of the injection of local anesthetic to the first bolus dose infused via a PCA device. Data regarding the use of PCA devices were collected using AccuLinker software (data extraction program of Accumate^®^1200 version 1.1, Woo Young Meditech, Seoul, Korea), which records the exact time and dose of every administration made by the device.

Secondary outcomes included: (1) duration of sensory and motor block, (2) 24-h cumulative opioid consumption, (3) pain severity (maximum [NRS], minimum [NRS], and frequency of severe pain) over 24 h, (4) sleep quality, (5) satisfaction score, (6) pre- and postoperative blood glucose concentrations, (7) hemodynamic measurements (systolic and diastolic blood pressure, heart rate), and (8) level of sedation. Patients were instructed to assess the sensory and motor functions of their blocked arm, compared with the contralateral arm or baseline (before the blockade), every 30 min postoperatively. The durations of sensory and motor blockade were defined as the times from the end of injection until the patient detected complete resolution of the sensory and motor blockades (end of self-assessment), respectively. The frequency of severe pain for 24 h postoperatively was reported using a 10-point scale, ranging from 0 for no severe pain to 10 for the consistent perception of severe pain. Sleep quality and satisfaction scores were assessed the morning after surgery on 10-point scales by a researcher, who was blinded to the group assignment. Preoperative and postoperative blood glucose concentrations on the day of surgery after transfer to the ward were measured using standard laboratory tests. Hemodynamic changes, including systolic blood pressure and heart rate, were recorded at seven time points from the start of sedation to 30 min after the end of sedative infusion (i.e., 0, 5, 10, and 30 min after the start of sedation; and 0, 10, and 30 min after the end of sedative infusion). Intraoperative sedation level was evaluated using the modified Ramsay Sedation Scale (mRSS) 5 and 10 min after the start of sedation and every 10 min thereafter through 50 min. After surgery, all patients transferred post anesthesia care unit and Modified Aldrete post-anesthesia score was adopted as the discharge criteria, which a score > 9 is needed for discharge. Patients who fulfilled the discharge criteria were transferred to the ward unit.

### 2.4. Statistical Analysis

Sample size was estimated based on our previous study, which evaluated the analgesic duration of BPB after sedation using dexmedetomidine [11]. That study showed that the mean ± standard deviation (SD) duration of analgesia in the dexmedetomidine group was 616.9 ± 158.2 min. Based on a 30% increase in duration of analgesia being clinically significant, and assuming a power of 0.9 and a two-sided alpha of 0.0083 (0.05/6) for multiple comparisons of post hoc analysis when the difference in variance of four groups is significant, then the minimum sample size would be 25 patients. Potential patient drop-out and data loss indicated that the minimum sample size was 120 patients (30 patients per group).

Depending on the results of Shapiro–Wilk tests, continuous variables were reported as mean ± SD and compared by independent *t*-tests or as median (interquartile range [IQR]) and compared by Kruskal–Wallis tests. Categorical variables were reported as number (%) and compared by using χ^2^ or Fisher’s exact tests. Time-to-event outcomes such as duration of analgesia (primary outcome) and sensory and motor blockade were determined by Kaplan–Meier survival analysis and compared by log-rank tests, with pairwise comparisons with *p*-value adjustment performed using the Benjamini and Hochberg method. The effects of each two drugs and the interaction term between them were tested using mixed-effect models for the analgesic duration and the repeated measurements (e.g., systolic blood pressure, heart rate, sedation score). A two-tailed *p*-value < 0.05 was considered statistically significant. Only the complete cases (without missing data for the primary outcome) were involved in the entire analysis. All analyses were performed using R software version 4.0.3 (R Project for Statistical Computing, Vienna, Austria).

## 3. Results

### 3.1. Study Participants

Total 126 patients were assessed for eligibility. Six patients refused to participate and were thus excluded, whereas the remaining 120 patients were randomly assigned to one of the four groups (30 patients per group). Primary outcomes could not be assessed in nine patients, due to the loss of log records from the PCA device. The CONSORT flow diagram is shown in Figure 2. The baseline characteristics of the participants are shown in Table 1. None of the patients experienced BPB failure.

### 3.2. Outcomes

Median (IQR) times to first request for analgesia differed significantly among the Control (8.1 [6.2 to 11.6] hours), DMED (9.0 [8.1 to 11.3] hours); DEXA (10.7 [8.1 to 20.5] hours), and DMED-DEXA (13.2 [11.5 to 19.1] hours) groups (*p* < 0.001) (Figure 3). Pairwise comparisons showed significant prolongation of analgesia in the DEXA included groups compared with the non-DEXA included groups (DEXA vs. control, *p* = 0.045; DEXA vs. DMED, *p* = 0.045; DMED-DEXA vs. control, *p* < 0.001; DMED-DEXA vs. DMED, *p* < 0.001) (Table A1 in Appendix A). A mixed-effect model showed that dexamethasone was the only significant factor associated with the duration of analgesia (*p* < 0.001) and the interaction between dexamethasone and dexmedetomidine was not significant (*p* = 0.274) (Table A2).

There were no differences in sensory and motor block duration (Figure A1 and Figure A2). Postoperative opioid consumption and pain severity were significantly reduced, whereas sleep quality and patient satisfaction scores were significantly higher, in the DEXA and DMED-DEXA groups than in the other two groups (Table 2). Postoperative glucose was higher in the DEXA, DMED-DEXA groups, but the change (before and after surgery) was not statistically significant.

Repeated measurements, including hemodynamic variables and sedation scores, are summarized in Table A3 in Appendix A. The changes in mRSS over time did not differ significantly between the groups (*p* = 0.2) (Figure 4). Hemodynamic changes during and after surgery are depicted in Figure A3 and Figure A4. Systolic blood pressure decreased over time (*p* < 0.001), and the degree of decrease over time was significantly different between the groups (interaction between group and time, *p* < 0.001). Systolic blood pressure continued to decrease after drug discontinuation in the DMED and DMED-DEXA groups, but did not further decrease in the Control and DEXA groups. Heart rate also showed similar results with systolic blood pressure (time and interaction between group and time, *p* = 0.003 and <0.001, respectively).

## 4. Discussion

The results of the current study suggest that intravenous dexamethasone can prolong analgesia. While the longest median analgesia duration was with patients receiving intravenous dexamethasone and dexmedetomidine sedation, the results of pairwise comparisons between the study groups and the results of the mixed effect model revealed that the prolonged duration of analgesia was actually due to the use of dexamethasone itself, and that no additional benefit could be attributed to the concurrent use of dexmedetomidine for sedation. In other words, the presence of dexmedetomidine did not alter the effect of dexamethasone on the analgesic duration of the blockade (i.e., no significant interaction between dexmedetomidine and dexamethasone).

In our previous study, dexmedetomidine sedation (mean dose of 1.6 μg/kg) was found to significantly prolong the analgesic duration of the blockade about 3 h [11]. A study comparing three doses of intravenous dexmedetomidine (0.5, 1.0, and 2.0 μg/kg) found that the duration of analgesia was prolonged significantly only in patients receiving 2.0 μg/kg of dexmedetomidine, not in those receiving 0.5 and 1.0 μg/kg [19]. In this context, the non-significant effect of dexmedetomidine noted in the current study could be explained by the doses used in DMED and DMED-DEXA groups (mean 1.2 and 1.1 µg/kg, respectively). This inference is also in line with other studies which used less than 1 µg/kg of dexmedetomidine [20,21].

A recent randomized trial evaluating the effects of intravenous adjuncts on the analgesic duration of interscalene block in patients undergoing arthroscopic shoulder surgery reported results similar to ours [22]. That study also found that analgesic duration in patients receiving both dexmedetomidine and dexamethasone was no longer than that in patients receiving dexamethasone alone. Although the result was similar to the current study, the effect of the dexamethasone or dexmedetomidine itself could not be revealed within the study, as they omitted control group.

In contrast to our findings, a previous study showed that co-administration of intravenous dexamethasone and dexmedetomidine markedly prolonged the duration of analgesia for interscalene BPB, compared with dexamethasone alone, in patients undergoing arthroscopic shoulder surgery [23]. Unfortunately, these studies cannot be directly compared, primarily because of differences in the type of surgery. In addition, there were differences in primary outcomes. The primary outcome of the current study was the time required for the first bolus dose of opioid via a PCA device, whereas it was the time required for the first rescue analgesics in the presence of basal infusion of opioid via a PCA device in the previous study. In addition, patients in the previous study basally infused opioid via a PCA device, resulting in no need for rescue analgesics in 23% of patients in the dexamethasone group and 50% of patients receiving dexamethasone and dexmedetomidine. Although the more practical primary outcome remains unclear, we believe that the outcome used in the current study is more sensitive for comparisons of analgesic duration.

The other outcomes are also important. First, the longer analgesic effect in the DMED-DEXA than in the other groups may be associated with lower pain scores and opioid consumption and better sleep quality in this group. The median duration of analgesia in this group (13.17 h), however, suggests that a considerable proportion of the patients experience pain during postoperative sleep. Strategies are therefore needed to prevent rebound pain after regional anesthesia [4]. Second, there were no significant differences in motor blockade among the four groups. In contrast, previous studies reported that intravenous administration of dexamethasone prolonged motor blockade [24,25], whereas intravenous dexmedetomidine did not [13,19,25]. Recently published network meta-analysis reports prolongation of sensory and motor blocks, especially with dexamethasone [25]. Such difference may be attributed to the distinct analgesic protocol, difference in the block technique and the accompanying baseline quality of the blockade. Also, self-reported outcome, the duration of motor and sensory blockade, may differ from clinical assessments by physicians. Third, high dose of dexamethasone used in the current study was based on a previous study which showed increased analgesic duration of single-shot ISB with ropivacaine with IV dexamethasone [26]. In recent published meta-analysis, only 10 mg of IV dexamethasone significantly prolonged analgesic duration of peripheral nerve blocks not 4 and 8 mg [27]. The same results were also shown in the recently published RCT according to dexamethasone dose in the sciatic nerve block [28]. High dose of dexamethasone may cause hyperglycemia, but the increase of blood glucose level was not significant in this study. Thus, its use should not be discouraged simply because of the concern for hyperglycemia unless the glucose level was poorly controlled preoperatively.

The present study had several limitations. First, midazolam was used for sedation in both the Control and DEXA groups. Its use was inevitable for blinding purposes, as systemic midazolam can induce reliable sedation without additional analgesic effects. Patients, however, were likely unable to discern the sedative agent, therefore not affecting the primary study outcome. Also, the use of midazolam might have affected the incidence of PONV as it has an antiemetic effect. Second, the dose of dexmedetomidine may not have been sufficient to prolong the analgesic effect of the blockade. However, the dosage used in the current study may be closer to the typical dose administered for sedation during regional anesthesia and may therefore cause fewer dose-related adverse effects. Although the duration of analgesia may correlate with the doses of dexmedetomidine, high doses of the latter may be associated with unintended side effects, including bradycardia, hypotension, excessive sedation, and prolonged recovery [11].

## 5. Conclusions

In conclusion, intravenous dexamethasone prolonged the analgesia duration of supraclavicular BPB after orthopedic upper extremity surgery. The concurrent use of mild to moderate sedation dose of intravenous dexmedetomidine in addition to intravenous dexamethasone showed no additional benefit to the prolongation of analgesia.

## Figures and Tables

**Figure 1 jpm-11-01267-f001:**
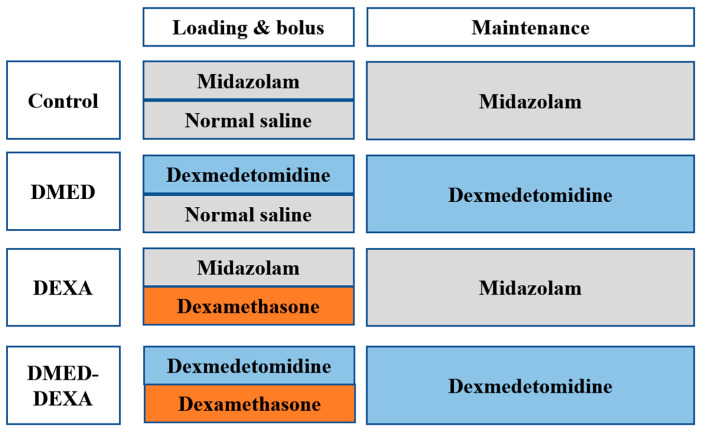
Schedule of drug administration to patients in the four study groups. Patients in the Control group received a 2 mL bolus of normal saline, followed by infusion of midazolam (0.05 mg/kg loading dose and a 20 µg/kg/h continuous dose). Patients in the DMED group received a 2 mL bolus of normal saline, followed by infusion of dexmedetomidine (1.0 µg/kg loading dose and a 0.4 µg/kg/h continuous dose). Patients in the DEXA group received a 2 mL bolus of dexamethasone, followed by infusion of midazolam. Patients in the DMED-DEXA group received a 2 mL bolus of dexamethasone, followed by infusion of dexmedetomidine.

**Figure 2 jpm-11-01267-f002:**
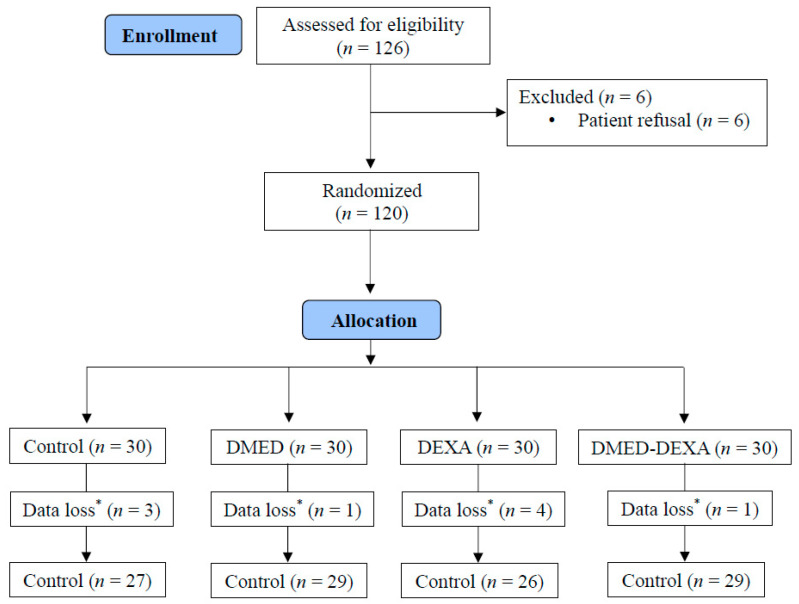
CONSORT flow diagram. * Data loss regarding the primary outcome (patient-controlled analgesia data).

**Figure 3 jpm-11-01267-f003:**
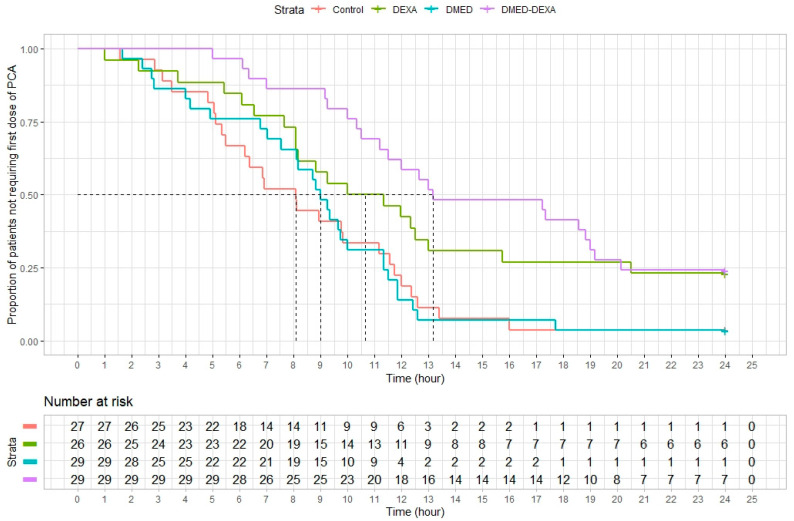
Survival analysis of the duration of analgesia. PCA, patient-controlled analgesia. The crossing points of the dotted and colored solid lines indicate median duration of the corresponding strata (groups). Median (IQR) times to first request for analgesia differed significantly among the Control (8.1 [6.2 to 11.6] hours), DMED (9.0 [8.1 to 11.3] hours); DEXA (10.7 [8.1 to 20.5] hours), and DMED-DEXA (13.2 [11.5 to 19.1] hours) groups (*p* < 0.001).

**Figure 4 jpm-11-01267-f004:**
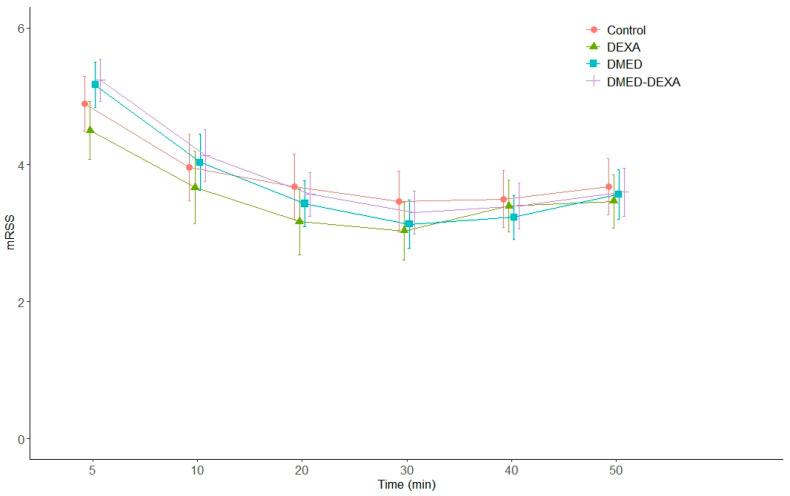
Changes in depth of sedation during and after surgery in the four study groups. mRSS (modified Ramsay Sedation Scale) 5 and 10 min after the start of sedation and every 10 min thereafter, up to 50 min.

**Table 1 jpm-11-01267-t001:** Demographic and clinical characteristics of the study groups. Results are reported as median [interquartile range (IQR)] or as number (%). * Duration of standard ASA monitoring during stay in the operating room.

	Control (*n* = 30)	DMED (*n* = 30)	DEXA (*n* = 30)	DMED-DEXA (*n* = 30)
Age (y)	57.5 [48.0–66.0]	57.5 [34.0–65.0]	55.5 [31.0–64.0]	55.5 [41.0–59.0]
Female, *n* (%)	13 (43.3%)	12 (40.0%)	15 (50.0%)	11 (36.7%)
Height (cm)	161.6 [154.0–170.0]	157.9 [150.0–171.6]	161.1 [156.0–170.0]	160.2 [154.9–166.5]
Weight (kg)	61.5 [56.4–68.0]	62.5 [54.7–71.0]	63.1 [55.8–76.0]	63.0 [53.7–69.0]
Preoperative blood glucose (mg/dL)	109.0 [96.0–119.0]	108.5 [97.0–117.0]	113.0 [100.0–128.0]	110.5 [96.0–133.0]
Monitoring time * (min)	81.0 [73.0–97.0]	99.0 [77.0–122.0]	81.0 [74.0–104.0]	91.5 [76.0–105.0]
Dexmedetomidine (µg)		75.6 [64.0–90.4]		67.8 [61.2–83.6]
Dexmedetomidine (µg/kg)		1.2 ± 0.3		1.1 ± 0.3
Midazolam (mg)	3.7 [3.1–4.3]		4.0 [3.5–4.7]	
Diagnosis, *n* (%)				
Fracture	27 (90.0%)	21 (70.0%)	21 (70.0%)	20 (66.7%)
Ulna impaction syndrome	3 (10.0%)	9 (30.0%)	9 (30.0%)	10 (33.3%)

**Table 2 jpm-11-01267-t002:** Secondary outcomes. Data are presented as median [IQR], mean ± SD or number (%). * Pain during the first 24 h postoperatively rated on a 10-point scale, from 0 for no severe pain to 10 for consistent perception of severe. NRS, numeric rating scale; PONV, post-operative nausea and vomiting.

	Control	DMED	DEXA	DMED-DEXA	*p*
	(*n* = 27)	(*n* = 29)	(*n* = 26)	(*n* = 29)	
● 24 h opioid consumption (μg)	225.0 [89.0–300.0]	250.0 [100.0–345.0]	150.0 [20.0–240.0]	127.5 [40.0–200.0]	0.044
● Minimal pain score (NRS)	3.0 [1.5–4.0]	3.0 [1.0–6.0]	1.5 [1.0–3.0]	1.5 [1.0–3.5]	0.104
● Maximal pain score (NRS)	7.2 ± 3.1	7.5 ± 2.4	5.1 ± 2.7	5.3 ± 3.2	0.004
● Frequency of the severe pain *	5.5 ± 2.7	6.0 ± 2.8	4.2 ± 2.9	4.2 ± 3.0	0.052
● Sleep quality	5.0 [1.0–10.0]	4.0 [0.0–7.0]	5.5 [3.0–10.0]	8.0 [5.5–10.0]	0.045
● Patient satisfaction	7.0 [5.0–10.0]	8.0 [5.0–10.0]	8.0 [6.0–10.0]	10.0 [7.5–10.0]	0.169
● Postoperative glucose (mg/dL)	109.0 [92.0–121.0]	105.0 [91.0–132.0]	113.0 [105.0–172.0]	117.0 [110.0–133.0]	0.032
● Glucose change (mg/dL)	−2.0 [−12.0–12.5]	−3.0 [−17.0–20.0]	11.5 [−9.0–47.0]	7.0 [−17.0–30.0]	0.547
● Use of rescue analgesics, *n* (%)	4 (14.8%)	5 (17.2%)	1 (3.8%)	1 (3.4%)	0.179
● PONV, *n* (%)	9 (33.3%)	13 (44.8%)	7 (26.9%)	10 (34.5%)	0.571

## Data Availability

The data presented in this study are available on request from the corresponding author.

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
