# Peer review of "The Effect of Intravenous Dexamethasone and Dexmedetomidine on Analgesia Duration of Supraclavicular Brachial Plexus Block: A Randomized, Four-Arm, Triple-Blinded, Placebo-Controlled Trial"

_jpm, 2021, doi:10.3390/jpm11121267_

Round 1
Reviewer 1 Report
Comments:
line 45 provide citation for anaestesia : Randomized, comparative study of the effectiveness of three different techniques of interscalene brachial plexus block using 0.5% ropivacaine for shoulder arthroscopy. Anaesthesiol Intensive Ther. 2017;49(1):47-52. doi: 10.5603/AIT.2017.0009
and analgesia : Randomised controlled trial of analgesic effectiveness of three different techniques of single-shot interscalene brachial plexus block using 20 mL of 0.5% ropivacaine for shoulder arthroscopy. Anaesthesiol Intensive Ther. 2017;49(3):215-221. doi: 10.5603/AIT.a2017.0031
line 112 - how many years of experience with how many blocks performed a year ?
line 117 whY Lovett's scale was not used to be more precise?
line 132 - how were the patients trainerd to ask for rescue postoperative analgesia - was NRS scale used? when where the patients trained to ask for first analgesia? if they experinnced aute? moderate? or any pain?
table 2 last line - if you analyse PONV, you should study the probability of incidence of PONV likewise in the study of Adverse Events during Vitrectomy under Adequacy of Anesthesia-An Additional Report. J Clin Med. 2021 Sep 15;10(18):4172. doi: 10.3390/jcm10184172
to ensure the homogeinity of the groups because PONV depends on the patients characteristics that you do not study - incude in the anthropometric data also these characteristics desribed in the Apfel score and discuss the rate of incidence of PONV in relation to the probability of its occurrence.
p 303 MIdazolam has aniemetic properties that may influence the rate of incidence of PONV
The Effect of Intravenous Midazolam on Postoperative Nausea and Vomiting: A Meta-Analysis. Anesth Analg. 2016 Mar;122(3):656-663. doi: 10.1213/ANE.0000000000000941
Author Response
Thank you for the opportunity to submit a revised version of our manuscript. The manuscript has been revised according to the reviewer's comments. All changes have been marked in red.

Reviewer 2 Report
I would firstly like the thank the authors for this clinically important, scientifically sound and well-presented research paper!
Indeed, as reference by the paper, there have been two other studies looking at the effect of this combination on regional block outcomes. However, the mixed-effect analysis and sound design of this paper give it significant merit.
It is curious that sensory and motor block duration did not mirror the extension seen in other studies with dexamethasone.
Group abbreviation required re-reading a few times. I wonder if DMED and DEXA is clearer?
Line 49 – perineural rather than intraneural
Line 125 - ?esomeprazole
Line 146 – To check – this is a patient reported score?
Line 148 – again unclear if this is rated by the researcher of self-reported by patient (presumably self-reported from later discussion)
Table 2 – seems like a high percentage for PONV in a population receiving sedation only and also an antiemetic (dexamethasone) in half the cases
Table 2 – are you able to mention the reason for missing data
Figure 3 – you have said ‘median sensory block duration’. Do you mean analgesic duration
Line 279 – Working in a country where the use of basal rate opioid infusion is uncommon. I would find this primary outcome to be more able to be extended to our practice and agree with your statement regarding sensitivity.
Line 281- was may – error
Line 282 – repeat DEXM-DEXA group
Line 284 - Given that patients may still experience pain during the night, and we are reporting on time to analgesia, and block duration. Are these findings influencing patient-centred metrics like patient satisfaction?
Author Response

(The authors gave the same response as above.)

Round 2
Reviewer 1 Report
Dear Authors,
congratulations on a good piece of scientific job.
I wish you good luck with your manuscript
Best regards.